# Validity of the Addiction-like Eating Behavior Scale among Patients with Compulsive Eating

**DOI:** 10.3390/nu16172932

**Published:** 2024-09-02

**Authors:** Camille Bourque, Maxime Legendre, Sylvain Iceta, Catherine Bégin

**Affiliations:** 1Research Center of the Quebec Heart and Lung Institute, Quebec City, QC G1V 4G5, Canada; camille.bourque@criucpq.ulaval.ca (C.B.); sylvain.iceta.1@ulaval.ca (S.I.); 2School of Psychology, Laval University, Quebec City, QC G1V 0A6, Canada; maxime.legendre.1@ulaval.ca; 3Centre d’Expertise Poids, Image et Alimentation (CEPIA), Laval University, Quebec City, QC G1V 0A6, Canada; 4Department of Psychiatry and Neurosciences, Laval University, Quebec City, QC G1V 0A6, Canada

**Keywords:** addictive eating, binge eating, compulsive eating, food addiction, obesity, validation

## Abstract

Food addiction (FA) and binge eating disorder (BED) co-occur and share compulsive eating symptoms. When using an FA measure, it is important to evaluate its performance in a population presenting compulsive eating. The study aims to validate the Addiction-like Eating Behavior Scale (AEBS) among a clinical sample characterized by compulsive eating and overweight/obesity and to evaluate its incremental validity over the Yale Food Addiction Scale 2.0 (YFAS). Patients seeking help for compulsive eating (n = 220), between January 2020 and July 2023, completed online questionnaires, including FA, compulsive eating, and BMI evaluations. The factor structure, internal consistency, and convergent, divergent, and incremental validity were tested. The sample had a mean age of 44.4 years old (SD = 12.7) and a mean BMI of 38.2 (SD = 8.0). The two-factor structure provided a good fit for the data, with factor loadings from 0.55 to 0.82 (except for item 15) and the internal consistency was high (ω = 0.84–0.89). The AEBS was positively correlated with the YFAS (r = 0.66), binge eating (r = 0.67), grazing (r = 0.47), craving (r = 0.74), and BMI (r = 0.26), and negatively correlated with dietary restraint (r = −0.37), supporting good convergent and divergent validity. For each measure of compulsive eating, linear regression showed that the AEBS “appetite drive” subscale had a unique contribution over the YFAS. This study provided evidence that the AEBS is a valid measure among a clinical sample of patients with compulsive eating and overweight/obesity. However, questions remain as to whether the AEBS is a measure of FA or compulsive eating.

## 1. Introduction

The high rate of obesity and its related consequences have challenged the scientific community for decades [1]. To provide effective solutions to manage the obesity epidemic, a much more advanced comprehension of the contributing factors is required [2,3]. For a subgroup of the population encountered in clinical settings, compulsive eating would be one of those key factors [4,5]. Compulsive eating has been associated with many physical (e.g., chronic diabetes, hypertension, and chronic headaches) and psychological (e.g., mood, anxiety, and sleep problems) consequences independent of weight status [6]. This relatively broad construct encompasses different specific eating behaviors or responses (e.g., grazing, binge eating, and emotional eating) that all have in common loss of control over food. For example, grazing involves repetitious and unplanned eating of small amounts of food throughout the day, while binge eating involves eating much larger than usual amounts of food in a restricted period, and emotional eating involves eating mostly in response to negative emotions. Recent studies have organized these behaviors into different dimensions [7,8] or on a continuum of severity [9,10]. Of particular interest, food addiction (FA), a form of compulsive eating that involves irresistible cravings and an inability to resist highly palatable foods, has received a lot of attention in the last two decades. FA has been theorized to be the most severe form of compulsive eating, that is a more severe and consolidated form of compulsion derived from repeated binge eating behaviors [9,10]. While some results support this hypothesis [11], the extensive similarities with the binge eating disorder (BED) have fueled a debate about the conceptualization of FA [12,13]. Nevertheless, many researchers believe that recognizing patients as addicted to food could lead to important public health implications like specific treatment development and accessibility, better regulation of ultra-processed foods, calorie-dense food taxes, and neutral food labeling [14].

The first efforts to conceptualize FA were made by adapting the diagnostic criteria for substance-related and addictive disorders (SRAD) to the area of eating difficulties [15,16]. In this conceptualization, palatable food was considered highly addictive like alcohol and drugs, explaining why some people could not resist eating it [17,18,19]. The well-known and widely used Yale Food Addiction Scale (YFAS), published by Gearhardt and colleagues [20], stems from this conceptualization and is based on the Diagnostic Statistical Manual (DSM-5) criteria for SRAD [21]. Despite the important contribution of the YFAS to the FA literature, the underlying conceptualization of the instrument remains disputed [22,23]. The main concern is whether FA should be treated in the same way as substance addiction considering fundamental differences between food and other addictive substances. Among the most frequently raised criticisms are that food is essential for survival (unlike substances), the magnitude of the addictive potential of food is not as significant as that of alcohol or drugs, and certain SRAD criteria may not apply perfectly to food [24,25].

In response to these criticisms, an alternative FA conceptualization was proposed, emphasizing eating behaviors rather than the addictive potential of food [26]. Based on the dual-process theories of motivation, which stipulate that addiction can develop from an imbalance between two systems, the automatic (impulsive system) and a regulatory executive system (reflexive system), Ruddock and colleagues [26] developed the Addiction-like Eating Behavior Scale (AEBS). The AEBS items originate from a qualitative study conducted among 210 participants from the university community, resulting in the generation of six characteristics and a pool of 62 items associated with FA [27]. A validation study was then carried out among a community sample of 555 individuals, reducing the number of items to 15, divided into two subscales: (1) appetite drive, which reflects increased reactivity to reward-related cues; and (2) low dietary control, which reflects decreased ability to exert inhibitory control [26]. Results of this study indicated a two-factor structure, good internal consistency and test–retest reliability, and convergent validity with measures of FA, binge eating, and emotional eating as well as body mass index (BMI). Of particular interest, the AEBS explained a part of BMI variance over and above the YFAS and the Binge Eating Scale (BES). To date, the AEBS has been translated into and validated in French [28], Italian [29], Portuguese [30], Turkey [31], and Chinese [32] but only two of these studies included clinical samples of patients with obesity [28,29]. These studies showed good convergent validity with moderate to strong correlations between the AEBS and the number of YFAS symptoms (r = 0.41–0.69), the severity of binge eating (r = 0.67–0.76), and BMI (r = 0.23–0.37) and good discriminant validity with small significant negative correlations between the AEBS “low dietary control” subscale and dietary restraint or restrained eating (r = −0.17–−0.27) [28,29]. The AEBS factorial structure has been tested only once with a non-community sample, using a sample of 502 inpatients with a BMI ≥ 35 km/m^2^, showing a good fit for the data with the two-factor structure [29].

Despite the good psychometric properties of the AEBS, Vanik and Meule [33] published a comment that cautioned researchers in the field of compulsive eating to create measures that correlate strongly with existing ones, thus contributing to the jangle fallacy in the field. To date, only two studies have examined the distinctive contribution of the AEBS in explaining FA and binge eating using clinical samples of patients with overweight or obesity [28,29]. Rossi and colleagues [29] evaluated the ability of the AEBS to differentiate between individuals with and without FA and individuals with and without BED, using the Receiver Operating Characteristics (ROC) curve analysis. They found that a cut-off point of ≥39 on the AEBS was highly accurate in distinguishing both groups, with 72% of correct classification for FA (according to the YFAS) and 78.5% for BED (according to the BES). Legendre and Bégin [28] obtained 77.3% of correct classification for FA (according to the YFAS) with a cut-off point of ≥41. They also found that both the AEBS and the YFAS significantly explained 8% and 9%, respectively, of depressive symptoms variance and neither significantly explained BMI variance. While the studies conducted among obesity samples are informative and interesting in terms of the psychometric properties of the AEBS, more studies are needed to examine the incremental validity of the AEBS over the YFAS with a sample of patients presenting compulsive eating.

The purpose of the study is twofold: (1) to validate the French-Canadian version of the AEBS among a clinical sample of patients seeking help for compulsive eating with overweight or obesity and (2) to evaluate the incremental validity of the AEBS over the YFAS in capturing compulsive eating and BMI. We hypothesize that the AEBS will demonstrate good psychometric properties (factor structure, internal consistency, and convergent/discriminant validity) similar to those found with community samples and patients with obesity. Additionally, we expect that the AEBS will capture a portion of compulsive eating variance that is not accounted for by the YFAS.

## 2. Materials and Methods

### 2.1. Participants and Procedure

The sample included 220 participants recruited through a multidisciplinary clinic, le Centre d’Expertise Poids, Image et Alimentation (Expertise Center for Weight, Body Image, and Eating Behaviors) between January 2020 and July 2023. The clinic offers care for individuals with eating disorders and weight management difficulties. To be included in the study, participants had to be over 18 years old, have a BMI of at least 25 kg/m^2^, and be seeking psychological assistance from the clinic. The mean BMI was 38.2 (SD = 8.0, range 35.3–70.4), with 85.3% reaching the obesity threshold and 14.7% reaching only the overweight threshold. The mean age was 44.4 years old (SD = 12.7, range 18–72). The sample was predominantly white (97.7%), female (89.5%), and employed full-time or part-time (73.6%). The remaining participants included retirees (14.5%), students (7.7%), and those unavailable for work (4.1%). More than 50% of participants had a university degree and an annual family income of 60,000 CAD or more.

To enroll in the study, participants completed questionnaires on LimeSurvey and participated in a short semi-structured interview with a psychologist to assess the presence of an eating disorder according to the DSM-5 criteria. Height and weight were self-reported to calculate BMI (kg/m^2^). This study was approved by the Laval University Research Ethics Committee (2018-205 CG R-4/14 April 2023) and was conducted in accordance with the principles of the Declaration of Helsinki.

### 2.2. Measures

#### 2.2.1. Food Addiction (Behavioral Approach)

The Addiction-like Eating Behavior Scale (AEBS) is a self-report questionnaire that measures eating behaviors related to FA based on 15 items [26], which are rated on a Likert scale from 1 (strongly disagree or never) to 5 (strongly agree or always). The instrument had been translated into French in 2020 [28]. The total score for all items ranged between 15 and 75 with a higher score corresponding to a more important endorsement of addiction-like eating behaviors. The AEBS can also be divided into two subscales: the “appetite drive” covers nine items with a score from 9 to 45 (e.g., “I continue to eat despite feeling full”, “I binge eat”) and the “low dietary control” includes six items with a score from 6 to 30 (e.g., “Despite trying to eat healthy, I end up eating ‘naughty’ foods”). Along with the good psychometric properties (internal consistency, divergent and convergent validity) of the AEBS, previous studies have confirmed invariance of the factor structure between sex in the general population [32] and between inpatients with severe obesity and individuals from the general population [29]. This suggests similar factor structure, an equivalent association of each item to the latent factor, and the same expected item response at the same absolute level of the trait.

#### 2.2.2. Food Addiction (Substance Approach)

The Yale Food Addiction Scale 2.0 (YFAS) relies on DSM-5 SRAD criteria to assess the symptoms of FA [20]. It is a self-report questionnaire containing 35 items, each answered on a Likert scale from 0 (never) to 7 (every day). Items cover 12 different dimensions of SRAD (eleven symptom criteria and one significant distress/functional impairment criterion) as consuming more than planned (e.g., “I continued to eat certain foods even though I was no longer hungry”), using despite consequences (e.g., “I kept eating in the same way even though my eating caused emotional problems”), craving (e.g., “I had such strong urges to eat certain foods that I couldn’t think of anything else”), or withdrawal (e.g., “When I cut down on or stopped eating certain foods, I felt irritable, nervous or sad”). The questionnaire can be used as a dichotomous diagnosis tool. To conclude regarding the presence of FA, at least two of the eleven symptom criteria must be met, in addition to the significant distress or functional impairment criterion (e.g., “I had significant problems in my life because of food and eating. These may have been problems with my daily routine, work, school, friends, family, or health”). The questionnaire can also be used to assess the severity of FA by summing up all the eleven symptom criteria, ranging from 0 to 11. The instrument had been previously translated and validated into French [34]. The internal consistency of the YFAS for the present study was high with a McDonald’s Omega of ω = 0.85.

#### 2.2.3. Binge Eating

The Binge Eating Scale (BES) is designed to assess symptoms associated with binge eating episodes [35]. It is a 16-item self-report questionnaire where participants have to choose from four statements the one that best describes their situation (e.g., “I have regular periods during the month when I eat large amounts of food, either at mealtime or at snacks”, “After I overeat, occasionally I feel guilt or self-hate”, or “I have days when I can’t seem to think about anything else but food”). Each statement has a value, ranging from zero to three. The sum of the score ranges from 0 to 46, and the highest score corresponds to more severe binge eating. A score of 17 or less indicates the presence of few or no episodes of binge eating, a score of 18–26 indicates moderate severity or frequency of binge eating, and a score of 27 or more indicates severe binge eating or a high frequency of episodes. The instrument had been previously translated and validated into French [36]. The internal consistency of the BES for the present study was high with a McDonald’s Omega of ω = 0.85.

#### 2.2.4. Grazing

The Grazing Questionnaire (GQ) measures behaviors and cognitions specific to grazing with loss of control [37]. It is a seven-item self-report questionnaire (e.g., “Do you find yourself picking at or nibbling food continuously?”) rated on a Likert scale from 0 (never) to 4 (always). All seven items can be summed up for a total score ranging from 0 to 28. The GQ showed high internal consistency (α = 0.82), good test–retest reliability (r = 0.67), and convergent validity with other measures of compulsive eating [37]. The instrument had been previously translated into French and used by our team [38]. The internal consistency of the GQ for the present study was high with a McDonald’s Omega of ω = 0.87.

#### 2.2.5. Craving

The Food Cravings Questionnaire Trait reduced (FCQTr) assesses affective, cognitive, and behavioral aspects of food craving as a trait [39]. It is a 15-item self-report questionnaire (e.g., “I feel like I have food on my mind all the time”, “I have no will power to resist my food crave?”) rated on a Likert scale from 1 (never or not applicable) to 6 (always). A higher total score represents a more intense food craving trait. A previous study validated the French version and showed a high internal consistency (α = 0.95) and a positive correlation with BMI (r = 0.30) and the number of FA symptoms (r = 0.71) [40]. The internal consistency of the FCQTr for the present study was high with a McDonald’s Omega of ω = 0.93.

#### 2.2.6. Dietary Restraint

The Three-Factor Eating Questionnaire (TFEQ) is designed to assess three aspects of eating: dietary restraint, disinhibition, and susceptibility to hunger [41]. It contains a total of 51 questions and is divided into two parts. The first part of the questionnaire includes 36 items with true or false answers whereas the second part consists of 15 items rated on a Likert scale from 1 to 4. In the present study, only the dietary restraint subscale was used to capture cognitive food restriction as well as restricted food intake to control body weight. This specific subscale contains 21 items with a score ranging from 0 to 21 (e.g., “I consciously hold back at meals in order not to gain weight”, “I do not eat some foods because they make me fat”). The instrument had been previously translated into and validated in French [42]. For the present study, the internal consistency was good with a McDonald’s Omega of ω = 0.80.

#### 2.2.7. Body Mass Index (BMI)

A sociodemographic questionnaire was used to collect data from participants: age, gender, ethnicity, education, marital status, household income, height, and weight. BMI was calculated based on standard procedure (kg/m^2^) with self-reported height and weight.

### 2.3. Statistical Analysis

Analyses were conducted using IBM SPSS 24.0 and Mplus 8.9 statistical software. Before proceeding with the analyses, the distributions of all variables were examined. To achieve the first objective, several analyses were completed. First, a confirmatory factor analysis (CFA) was performed to ensure the reproduction of the two-factor structure with our sample. The following indices were used to measure data adequacy: normed χ^2^ statistic (χ^2^/df), comparative fit index (CFI), root-mean-square error of approximation (RMSEA), and normalized root-mean-square residual (SRMR). An χ^2^/df ratio of 3 or less and a value greater than 0.90 for CFI indicate an acceptable fit, while values of 0.08 or less are acceptable for RMSEA and SRMR [43,44]. Also, McDonald’s Omega was used to assess the internal consistency of the scale, with 0.70 considered the acceptable lower limit [45]. The McDonald’s Omega operates with more flexible conditions and provides more accurate estimates [46]. Finally, correlations were performed between the AEBS and the YFAS, the BES, the GQ, the FCQTr, and BMI to assess convergent validity, and between the AEBS and the TFEQ dietary restraint to assess divergent validity. To evaluate the incremental validity of the AEBS in capturing compulsive eating and additive behaviors, stepwise linear regressions were performed using both AEBS subscales and the YFAS symptoms count to predict the BES, the GQ, the FCQTr, and BMI.

## 3. Results

### 3.1. Factorial Structure and Internal Consistency

The two-factor structure was confirmed and provided a good fit for the data (normed χ^2^ (χ^2^/df) = 1.45, CFI = 0.973, RMSEA (90% CI) = 0.045 (0.026–0.062), SRMR = 0.054). Covariance pathways between error terms were added between items 8–10, 11–12, and 2-3-4-9 to improve the fit of the data. Factor loadings were all significant and ranged from 0.55 to 0.82, except for item 15, which had a factor loading of 0.27 (Table 1). McDonald’s Omega revealed high internal consistency for the entire questionnaire (ω = 0.89) and for both subscales, “appetite drive” (ω = 0.84) and “low dietary control” (ω = 0.88).

### 3.2. Convergent and Discriminant Validity

Descriptive statistics are presented in Table 2. Each measure of FA (AEBS and YFAS) as well as measures of compulsive eating (BES, GQ, and FCQTr) were strongly correlated with each other, with coefficients ranging from 0.46 to 0.74 (Table 3). The AEBS “appetite drive” subscale showed high coefficients with compulsive eating scales (r from 0.49 to 0.72), while the AEBS “low dietary control” showed smaller coefficients with compulsive eating scales (r from 0.33 to 0.57). The correlations between BMI and measures of FA were small (r from 0.19 to 0.33), and there were no significant correlations between BMI and any measures of compulsive eating. Finally, dietary restraint showed small negative correlations with measures of FA (r from −0.25 to −0.37), but there were no significant correlations between dietary restraint and measures of compulsive eating.

### 3.3. Incremental Validity

Using both AEBS subscales and the YFAS symptoms count, the stepwise regression models were able to explain 52% of binge eating, 27% of grazing, 59% of cravings, and 10% of BMI (Table 4). For each measure of compulsive eating, the AEBS “appetite drive” subscale had the strongest unique contribution, while the YFAS symptoms count also made a significant unique contribution, though smaller than the AEBS. The AEBS “low dietary control” had a unique contribution only for cravings. For BMI, only the YFAS symptoms count had a significant unique contribution.

## 4. Discussion

This study provided evidence that the AEBS is a valid measure among a clinical sample of patients with compulsive eating and overweight/obesity, increasing confidence in its use with clinical populations experiencing eating difficulties and eating disorders. The two-factor structure was replicated with good internal consistency, supporting the applicability of the two-dimensional model of addiction for this population [26]. The AEBS total score and both subscales showed moderate to strong correlations with FA symptoms and binge eating, replicating previous results regarding convergent validity with clinical samples with obesity [28,29] while extending convergent validity to other dimensions of compulsive eating, such as grazing and cravings. In sum, a higher score on the AEBS was associated with a higher score on every compulsive eating measure. In addition, small negative correlations between the AEBS total score, both AEBS subscales, and dietary restraint suggest good divergent validity. Previous studies had also reported negative correlations with dietary restraint [26,28] or restrained eating [29] particularly with the AEBS “low dietary control”. Another important finding was that the pattern and strength of correlations observed between the two FA measures, the YFAS and the AEBS, and all measures of compulsive eating were essentially similar. This suggests that both FA measures perform quite similarly in relation with compulsive eating measures.

The incremental validity of the AEBS over the YFAS aimed to clarify its relevance in explaining compulsive eating. Using three measures of compulsive eating (binge eating, grazing, and craving), it was revealed that the “appetite drive” subscale was particularly effective in explaining compulsive eating variance, while the “low dietary restraint” subscale provided a very limited unique contribution. The items in this subscale mostly cover food choices and diet quality, which might translate into a nutritional dimension of FA, potentially explaining why this subscale was less effective in accounting for compulsive eating. The “appetite drive” subscale offered a better explanation of variance than the YFAS symptoms count for every measure of compulsive eating. These results suggest good incremental validity for the AEBS, with a small additional amount of variance explained by the YFAS. However, the high proportion of explained variance of the “appetite drive” subscale might be boosted by items that are not FA specific, such as “I binge eat” (Appendix A), which is closely related to compulsive eating. This item was the strongest predictor of binge eating, grazing, and craving, explaining more than half the variance for these three variables on its own. This leads to the conclusion that the AEBS might be closer to a measure of compulsive eating rather than a measure of FA, especially since the AEBS is not based on established criteria for substance or behavioral addiction. Finally, the AEBS subscales failed to explain BMI variance, whereas the YFAS symptoms count succeeded in explaining it. This finding contrasts with the original study with a community sample, which showed a small portion of BMI variance explained by the AEBS while controlling for the YFAS and the BES [26]. The discrepancy may be due to sample differences, specifically the narrower BMI range in individuals with overweight or obesity compared to the general population. It is also important to note that many factors influence BMI over time, making it generally difficult to explain.

The main interest of the AEBS lies in its basis on a theoretical model targeting mechanisms of addiction, offering a different conceptualization from the YFAS. It is useful for capturing a greater severity of compulsive eating; however, it remains challenging to determine whether it measures a dimension that specifically represents FA. Despite being rooted in dual-process theories of motivation to explain addiction, empirical data suggest that the AEBS primarily measures eating behavior patterns more akin to compulsive eating rather than FA itself. Some studies have used the YFAS as a gold standard to determine whether the AEBS correctly classifies patients as having FA. AEBS cut-offs of ≥39 and ≥41 have been suggested to ensure good sensitivity for identifying patients likely to have FA [28,29]. However, these cut-offs have shown limited specificity in samples with compulsive eating, resulting in many false positives (Appendix A). A higher cut-off (e.g., ≥49) would achieve better specificity but at the cost of sensitivity. A significant challenge in establishing whether the AEBS properly measures FA is its reliance on the YFAS as the gold standard for FA. Although the YFAS is derived from SRAD criteria, it remains highly correlated with other measures of compulsive eating, complicating the distinction between FA and other forms of compulsive eating. Additionally, the YFAS requires individuals to acknowledge their distress, which is typically assessed by a qualified professional, potentially affecting the optimal classification of patients. To advance the field, developing a diagnostic interview to better assess FA and distinguish it from other forms of compulsive eating would be essential. Nevertheless, the AEBS remains a brief and valid measure of compulsive eating with a very short administration time, and efforts should focus on ensuring its unique empirical contribution to FA measurement.

The current study has limitations that should be considered. Firstly, BMI was calculated using self-reported height and weight. This may have had an influence on the analyses involving BMI, although likely a minor one. Secondly, a large proportion of the participants were women. While this reflects the demographics of patients encountered in clinical services, it is important to gather further evidence on the psychometric properties of the AEBS with a male clinical sample. Thirdly, the cross-sectional design of the study does not allow us to establish causality between FA and compulsive eating. It would have been valuable to examine how the presence of FA might predict compulsive eating prospectively. Therefore, the results of the linear regressions should be interpreted with caution, without implying a causal relationship. Finally, the invariance of the factor structure between BMI categories could not be tested due to the size of the sample and the low proportion of patients reaching only the overweight threshold.

## 5. Conclusions

This study provided evidence that the AEBS is a valid measure among a clinical sample of patients with compulsive eating and overweight or obesity. Notably, the AEBS “appetite drive” subscale offered a better explanation of variance than the YFAS symptoms count for every measure of compulsive eating. However, questions remain about what exactly is measured by the AEBS. Despite being based on a different theoretical framework, the AEBS remains highly correlated with the YFAS and other measures of compulsive eating, making it difficult to clearly identify its specificity over other measures. To avoid misinterpretation and to advance the field of FA, we suggest developing a semi-structured interview to assess FA and differentiate it from other dimensions of compulsive eating.

## Figures and Tables

**Table 1 nutrients-16-02932-t001:** Standardized factor loadings for each item.

Appetite Drive									Low Dietary Control					
1	2	3	4	5	7	9	14	15	6	8	10	11	12	13
0.72	0.67	0.66	0.63	0.82	0.66	0.55	0.63	0.27	0.61	0.74	0.71	0.71	0.71	0.81

**Table 2 nutrients-16-02932-t002:** Descriptive statistics.

Variables	Means (SD)
AEBS	50.3 (9.4)
AEBS—appetite drive	30.5 (5.7)
AEBS—low dietary control	19.9 (5.0)
YFAS symptoms	5.3 (3.2)
BES	23.6 (8.7)
GQ	15.6 (5.4)
FCQTr	56.5 (11.9)
TFEQ—dietary restraint	8.1 (4.3)
BMI	38.2 (8.0)

Note. AEBS = Addiction-like Eating Behavior Scale; YFAS = Yale Food Addiction Scale; BES = Binge Eating Scale; GQ = Grazing Questionnaire; FCQTr = Food Cravings Questionnaire Trait reduced; TFEQ = Three-Factor Eating Questionnaire; BMI = Body Mass Index.

**Table 3 nutrients-16-02932-t003:** Correlations between study variables.

	1	2	3	4	5	6	7	8	9
1. AEBS	1	0.90 *	0.87 *	0.66 *	0.67 *	0.47 *	0.74 *	−0.37 *	0.26 *
2. AEBS—appetite drive		1	0.55 *	0.61 *	0.68 *	0.49 *	0.72 *	−0.31 *	0.19 *
3. AEBS—low dietary control			1	0.54 *	0.49 *	0.33 *	0.57 *	−0.35 *	0.28 *
4. YFAS symptoms				1	0.61 *	0.46 *	0.64 *	−0.25 *	0.33 *
5. BES					1	0.60 *	0.72 *	−0.02	0.07
6. GQ						1	0.60 *	−0.12	−0.01
7. FCQTr							1	−0.15	−0.15
8. TFEQ—dietary restraint								1	−0.15
9. BMI									1

Note. AEBS = Addiction-like Eating Behavior Scale; YFAS = Yale Food Addiction Scale; BES = Binge Eating Scale; GQ = Grazing Questionnaire; FCQTr = Food Cravings Questionnaire Trait reduced; TFEQ = Three-Factor Eating Questionnaire; BMI = Body Mass Index. * *p* < 0.001.

**Table 4 nutrients-16-02932-t004:** Prediction of compulsive eating and BMI using AEBS subscales and YFAS symptoms.

	*β*	*t*	*F*	*R*	*R* ^2^	*R*^2^ adj.
BES score						
Step 1			187.90 *	0.68	0.46	0.46
AEBS—Appetite drive	0.68	13.71 *				
Step 2			118.62 *	0.72	0.52	0.52
AEBS—Appetite drive	0.49	8.29 *				
YFAS symptoms	0.31	5.19 *				
GQ score						
Step 1			67.27 *	0.49	0.24	0.23
AEBS—Appetite drive	0.49	8.20 *				
Step 2			42.08 *	0.53	0.28	0.27
AEBS—Appetite drive	0.32	4.44 *				
YFAS symptoms	0.26	3.63 *				
FCQTr score						
Step 1			232.26 *	0.72	0.52	0.51
AEBS—Appetite drive	0.72	15.24 *				
Step 2			147.96 *	0.76	0.58	0.57
AEBS—Appetite drive	0.53	9.43 *				
YFAS symptoms	0.31	5.60 *				
Step 3			106.86 *	0.77	0.60	0.59
AEBS—Appetite drive	0.46	7.99 *				
YFAS symptoms	0.25	4.43 *				
AEBS—Low dietary control	0.18	3.32 *				
BMI						
Step 1			17.63 *	0.33	0.11	0.10
YFAS symptoms	0.33	4.20 *				

Note. BMI = Body Mass Index; AEBS = Addiction-like Eating Behavior Scale; YFAS = Yale Food Addiction Scale; BES = Binge Eating Scale; GQ = Grazing Questionnaire; FCQTr = Food Cravings Questionnaire Trait reduced. * *p* < 0.001.

## Data Availability

The data supporting the conclusions of this article will be made available by the authors on reasonable request.

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
