# Peer review of "Validity of the Addiction-like Eating Behavior Scale among Patients with Compulsive Eating"

_nutrients, 2024, doi:10.3390/nu16172932_

Round 1

Reviewer 1 Report

Comments and Suggestions for Authors

Catherine Bégin et al. submitted to Nutrients an article, dealing with the validity of the addiction-like eating behavior scale among patients with FA and BED.

This manuscript appears well structured, but it requires some clarifications.

Authors must state when the study was conducted, both in the text and in the abstract.

Please, describe in the introduction the Public Health implications underlying the behaviors described and investigated, with the related main consequences and outcomes.

Comments on the Quality of English Language

 Minor editing of English language required.

Author Response

Catherine Bégin et al. submitted to Nutrients an article, dealing with the validity of the addiction-like eating behavior scale among patients with FA and BED. This manuscript appears well structured, but it requires some clarifications.

  1. Authors must state when the study was conducted, both in the text and in the abstract.

We have added the information in the abstract and in the method’s section (Participants and procedure).

  1. Please, describe in the introduction the Public Health implications underlying the behaviors described and investigated, with the related main consequences and outcomes.

There is a lot of literature on the subject, and it would be difficult to make an exhaustive list given the space available and the purpose of the article. Nevertheless, in the first paragraph, we have included some of the consequences of compulsive eating and some of the public health implications of recognizing food addiction.

Reviewer 2 Report

Comments and Suggestions for Authors

Dear Authors,

Thank you for submitting your manuscript. Please see my comments below:

  • Abstract, line 19: Could you clarify what you mean by "higher weight"? Also, sample characteristics should be provided, according to age and BMI.

  • In the first two paragraphs of the Introduction, it would be helpful to provide a clearer explanation of the differences between the concepts of compulsive eating, binge eating, and food addiction.

  • In Section 2.1, please include the age range of the study participants. Additionally, providing the mean BMI, as well as the percentage of participants with a BMI ≥ 25 kg/m² and ≥ 30 kg/m², would offer a more comprehensive characterization of your patient population.

  • Lines 121-122: Could you explain what you mean by "unfit for work" and rephrase the statement to be more clear?

  • Body mass index should be provided in the description of the methods section.
  • Methods and Results: The manuscript lacks information on the previous or current invariance testing of the AEBS, which is essential for establishing construct validity evidence.

Author Response

Dear Authors, thank you for submitting your manuscript. Please see my comments below:

  1. Abstract, line 19: Could you clarify what you mean by "higher weight"? Also, sample characteristics should be provided, according to age and BMI.

We have replaced “high weight” with “overweight/obesity”, which better reflects the study’s inclusion criteria (BMI ≥ 25). Also, we have added the sample characteristics in the abstract.

  1. In the first two paragraphs of the Introduction, it would be helpful to provide a clearer explanation of the differences between the concepts of compulsive eating, binge eating, and food addiction.

In the first paragraph, we have added examples to help readers understand the differences between compulsive eating, grazing, binge eating, emotional eating, and food addiction.

  1. In Section 2.1, please include the age range of the study participants. Additionally, providing the mean BMI, as well as the percentage of participants with a BMI ≥ 25 kg/m² and ≥ 30 kg/m², would offer a more comprehensive characterization of your patient population.

We have added the information in section 2.1.

  1. Lines 121-122: Could you explain what you mean by "unfit for work" and rephrase the statement to be more clear?

We have replaced "unfit for work" with "unavailable for work". These are participants who were unable to work during the recruitment period.

  1. Body mass index should be provided in the description of the methods section.

We have added the information in the section 2.2.7.

  1. Methods and Results: The manuscript lacks information on the previous or current invariance testing of the AEBS, which is essential for establishing construct validity evidence.

We have added information about previous invariance testing in the Materials and Methods section 2.2.1. In the present study, we only have a clinical sample of 220 participants which is insufficient for invariance testing between subgroups, like invariance testing according to BMI categories. We have added a sentence in the limitation section.
